# A Janus carbaporphyrin pseudo-dimer

Haodan He[1], Jiyeon Lee[2], Zhaohui Zong[1], Jiwon Kim [2,6], Vincent M. Lynch[3], Juwon Oh [4] ✉, Dongho Kim [5] ✉, Jonathan L. Sessler [3] ✉ & Xian-Sheng Ke [1] ✉

Carbaporphyrin dimers, investigated for their distinctive electronic structures and exceptional properties, have predominantly consisted of systems containing identical subunits. This study addresses the associated knowledge gap by focusing on asymmetric carbaporphyrin dimers with Janus-like characteristics. The synthesis of a Janus-type carbaporphyrin pseudo-dimer **5** is presented. It displays antiaromatic characteristics on the fused side and nonaromatic behavior on the unfused side. A newly synthesized tetraphenylene (TPE) linked bis-dibenzihomoporphyrin **8** and a previously reported dibenzo[g,p]chrysene (DBC) linked bis-dicarbacorrole **9** were prepared as controls. Comprehensive analyses, including ¹H NMR spectral studies, single crystal X-ray diffraction analyses, and DFT calculations, validate the mixed character of **5**. A further feature of the Janus pseudo-dimer **5** is that it may be transformed into a heterometallic complex, with one side coordinating a Cu(III) center and the other stabilizing a BODIPY complex. This disparate regiochemical reactivity underscores the potential of carbaporphyrin dimers as versatile frameworks, with electronic features and site-specific coordination chemistry controlled through asymmetry. These findings position carbaporphyrin dimers as promising candidates for advances in electronic structure studies, coordination chemistry, materials science, and beyond.

Carbaporphyrins, characterized by the replacement of one or more nitrogen atoms in the porphyrin core with a carbon atom, are intriguing due to their diverse aromaticity features and unique metal coordination properties[1]. Previous research has predominantly focused on monomeric carbaporphyrins, including benziporphyrin[2], azuliporphyrin[3], and N-confused porphyrins[4–6], among others[1,7,8]. Recently, attention has shifted towards carbaporphyrin dimers, prompting investigations into their distinct topologies, photophysical characteristics, potential applications in multi-metal coordination chemistry, and synergistic effects between subunits. While various symmetric systems have been reported, featuring identical halves in the dimer, there is a dearth of exploration into carbaporphyrin dimers with two distinct porphyrin-like subunits.

Notable examples of symmetric systems include N-confused porphyrin dimers by Furuta and colleagues[9–17], directly linked, metal-linked, and alkyl bridged N-confused porphyrin dimers by Chmielewski and coworkers[18–26], an adj-dicarbaporphyrin dimer with a tri-palladium sandwich complex by Lash and coworkers[27], azuliporphyrin dimers and a thia-carbaporphyrin dimer by Latos-Grażyński and coworkers[28–30], carbaporphyrin dimers created through the metalation of expanded porphyrins by Osuka and coworkers[31], benzonorrole and N-confused corrole dimers by Furuta and colleagues[15,16], a naphthene-linked thia-carbaporphyrin dimer by Cho and coworkers[32], a piperazine-bridged thianorrole dimer by Ravikanth and coworkers[33], a carbatriphyrin(3.1.1) dimer by Srinivasan and coworkers[34], a pyrene-linked carbaporphyrin dimer and an earring carbaporphyrin dimer

¹College of Chemistry, Beijing Normal University, Beijing 100875, China. ²School of Integrated Technology, College of Computing, Yonsei University, Incheon 21983, Korea. ³Department of Chemistry, The University of Texas at Austin, Austin, TX 78712-1224, USA. ⁴Department of Chemistry, Soonchunhyang University, Asan 31538, Korea. ⁵Department of Chemistry, Yonsei University, Seoul 03722, Korea. ⁶Present address: Integrated Science and Engineering Division, Underwood International College, Yonsei University, Incheon 21983, Korea. ✉e-mail: juwoh933@sch.ac.kr; dongho@yonsei.ac.kr; sessler@cm.utexas.edu; kexiansheng@bnu.edu.cn

**Fig. 1 | Chemical structures of 5, 8 and 9. (5)** Janus carbaporphyrin pseudo-dimer, **(8)** TPE-linked bis-dibenzihomoporphyrin, **(9)** DBC-linked bis-dicarbacorrole.

linked by a porphyrin backbone by Song and collaborators[35,36], and a neo-fused hexaphyrin dimer by Xie and coworkers[37]. Our group contributed to this body of work by reporting a dibenzo[g,p]chrysene linked carbaporphyrin dimer, demonstrating how the formation of a bimetallic complex could modify the electronic structure of the π-system as a whole[38,39].

Despite these achievements, the exploration of carbaporphyrin dimers with two distinct porphyrin-like subunits remains essentially uncharted territory[30,40,41]. In this study, we introduce a Janus carbaporphyrin pseudo-dimer 5 (Fig. 1), featuring two distinct sides that display antiaromatic and nonaromatic properties, respectively, along with different coordination chemistry properties.

Janus molecules represent a class of compounds characterized by the manifestation of distinct chemical or physical attributes on two distinct sides of the same molecular entity[42–45]. The resultant asymmetry in their molecular structure imparts unique assembly behaviors at the nanoscale, prompting extensive exploration of Janus-type systems within the realms of biotechnology and materials science[46–56]. Notably, the aggregation states of Janus constructs, encompassing dendrimers[46,53–56], nanoparticles[47,48,50,52], and metal-organic assemblies[44], have been subjects of considerable research attention. Conversely, scant attention has been devoted to investigations of their intrinsic electronic and photophysical properties at the molecular level.

To address this gap in knowledge, the current study was initiated. Specifically, we designed and successfully synthesized an asymmetric carbaporphyrin pseudo-dimer (compound 5). Subsequently, we conducted a systematic comparative analysis of its electronic structure and photophysical properties in relation to those of a newly synthesized tetraphenylene (TPE)-linked bis-dibenzihomoporphyrin, designated as compound 8, and a previously reported dibenzo[g,p] chrysene (DBC)-linked bis-dicarbacorrole, identified as compound 9 (cf. Fig. 1)[38]. This analytical approach was designed to provide an understanding of the distinctive characteristics inherent to these molecular entities.

As detailed in subsequent sections, the carbarporphyrin pseudo-dimer 5 manifests Janus-like chemical features, wherein the fused side (comprising a dicarbacorrole with a fused phenylene) exhibits antiaromaticity, while the unfused side (comprising a dibenzihomoporphyrin with an open phenylene) displays nonaromatic characteristics. This conclusion is supported by [1]H NMR spectroscopic studies and density functional theory (DFT) calculations. In contrast, the control dimers 8 and 9 exhibit nonaromatic and antiaromatic character, respectively, within their respective constituent segments.

The capacity of pseudo-dimer 5 to stabilize a mixed BF$_2$−Cu(III) complex, a phenomenon absent in controls 8 and 9, serves as prima facie evidence for the Janus-type character inherent in pseudo-dimer 5. This observation underscores the broader implications of creating desymmetrized macrocycles relative to more conventionally symmetric systems, an ability that is expected to translate into advances in our understanding of structure-function properties in porphyrin analogue chemistry while providing potential advantages in a diverse array of chemical applications where control over electronic structure or coordination chemistry is essential.

## Results

The synthesis of the Janus carbaporphyrin pseudo-dimer 5 is shown in Fig. 2a. 3,6-Dibromo-9,10-bis(4-bromophenyl)phenanthrene 2 was prepared by oxidizing tetra(p-bromophenyl)ethene 1 with one equivalent 2,3-dichloro-5,6-dicyano-p-benzoquinone (DDQ) in the presence of trifluoromethanesulfonic acid. Intermediate 2 was then reacted with mesitaldehyde in the presence of n-butyllithium to form the tetracarbinol 3. Heating 3 with excess pyrrole in the presence of BF$_3$·OEt$_2$ yielded the tetrapyrrole 4. A BF$_3$·OEt$_2$-catalyzed condensation of 4 with pentafluorobenzaldehyde, followed by oxidation with excess DDQ, then produced the Janus carbaporphyrin pseudo-dimer as a dark green solid 5 in 6% yield following purification by silica gel chromatography followed by gel permeation chromatography (GPC). The control carbaporphyrin dimer 8 was prepared using a similar route (Supplementary Fig. 2). Both 5 and 8 were characterized by [1]H, [19]F, 2D-correlation spectroscopy (COSY) NMR spectroscopies, UV-vis-NIR spectroscopy, high-resolution mass spectrometry, as well as single crystal X-ray diffraction analysis.

Diffraction grade single crystals of 5 and 8 were obtained through the slow diffusion of methanol into initial toluene solutions (Fig. 3). The resulting X-ray diffraction structure confirmed the asymmetric nature of 5, which was revealed to contain a dicarbacorrole subunit with fused phenylenes and a dibenzihomoporphyrin subunit with unfused phenylenes. The phenanthrene unit shows less distortion than in the previously reported bis-dicarbacorrole 9 containing double-fused phenylenes[38], a finding consistent with a lower level of strain due to mono-fusion of the phenylene subunits in 5. In contrast, the crystal structure of 8 revealed a highly symmetric conformation and the presence of two identical dibenzihomoporphyrin subunits. The C(5) − C(6) bond lengths of 8 were determined to be 1.362 Å, indicating significant double bond character. The corresponding bond length is 1.382 Å in the case of 5. This finding lends support to the conclusion that fusion of the phenylene subunits serves to reduce the double bond character of the C(5) − C(6) bond. Consistent with this

**Fig. 2 | Synthesis of 5, 7 and 10. a** Synthesis of **5** and its mixed BF$_2$−Cu(III) complex **7**. Conditions: (i) 1 equiv. DDQ and trifluoromethanesulfonic acid in dichloromethane (DCM) with stirring at 0 °C for 0.5 h, then room temperature for 12 h, 30%; (ii) *n*-BuLi in THF, −78 °C, 1 h, then mesitaldehyde, −78 °C, 1 h, 71%; (iii) pyrrole and BF$_3$·OEt$_2$, reflux, 20 h, 75%; (iv) pentafluorobenzaldehyde and BF$_3$·OEt$_2$ in dry DCM, room temperature, 2.5 h, followed by DDQ, room temperature, 0.5 h, 6%; (v), BF$_3$·OEt$_2$ and N,N-diisopropylethylamine (DIPEA) in toluene, with stirring at 80 °C for 12 h, 70%; (vi) Cu(OAc)$_2$ in CHCl$_3$/CH$_3$CN (v/v 3:2), reflux, 48 h, 24%. Mes and C$_6$F$_5$ denote mesityl and pentafluorophenyl groups, respectively; (**b**) Synthesis of bis-BODIPY complex **10**. Conditions: (vii), BF$_3$·OEt$_2$ and N,N-diisopropylethylamine (DIPEA) in toluene, with stirring at 80 °C for 12 h, 59%.

suggestion is the fact that the C(5) − C(6) bond length (1.392 Å) in the bis-dicarbacorrole **9** is yet longer still[38]. The crystal structures of **5** and **8** revealed C(10) − C(11), C(1) − C(21), C(10') − C(11') and C(1') − C(21') bond lengths in the 1.465 − 1.476 Å range (Fig. 3), indicating that the central arene moieties are bound to the dipyrromethene subunits through single bonds in both macrocycles. This finding is consistent with what was seen in the bis-dicarbacorrole **9** and other reported dicarbaporphyrinoids[57–59].

The ¹H NMR spectra of the free-base forms **5** and **8** were recorded in CD$_2$Cl$_2$ (Fig. 4). As might be anticipated given its less symmetric structure, the ¹H NMR spectrum of **5** is considerably more complicated than that of **8**. On the basis of 2D COSY and D$_2$O exchange experiments (Supplementary Figs. 20–23 and 33–35), the proton signals for **5** and **8** could be readily assigned and are provided in Fig. 4. In the case of **5**, there are two doublets located at 5.16 and 5.49 ppm, which are assigned to the pyrrolic *β*-Hs on the fused side. The inner CHs (H22 and H25) and NH (H26) at the fused side are found at 16.99 and 17.83 ppm, respectively. Thus, the average chemical shift differences between the

pyrrolic NH and *β*-H protons (Δ*δ*) and the inner CH and outer CH protons (Δ*δ'*) at the fused side were 12.5 and 10.8 ppm, respectively. These values are larger than seen in the bis-dicarbacorrole **9**[38] (Δ*δ* and Δ*δ'* are 10.3 and 7.8 ppm, respectively). In addition, Δ*δ'* is larger than that of the phenathriporphyrin[57] (Δ*δ'* is calculated as 10.3 ppm based on the average chemical shift differences between H(3,8),H(2,9) and inner CHs). These observations lead us to suggest an enhanced level of antiaromaticity on the fused side of pseudo-dimer **5** compared to the control system **9** and phenathriporphyrin. In contrast, the pyrrolic *β*-Hs on the unfused side of **5** resonate at 5.87 and 6.19 ppm, while the NH (H26') signal in this half is found at 12.30 ppm. Thus the average chemical shift differences between the pyrrolic NH and *β*-H protons (Δ*δ*) is 6.27 ppm for this portion of the pseudo-dimer, a value that is comparable to that for corresponding nonaromatic systems, such as biphenylcorrole[58] (4.64 ppm) and a carbaporphyrin cage[60] (5.98 ppm). We thus suggest that the unfused side of **5** is nonaromatic in character.

In the case of the TPE-linked bis-dibenzihomoporphyrin control **8**, the NH signal was found at 13.44 ppm (Fig. 4), which is comparable to

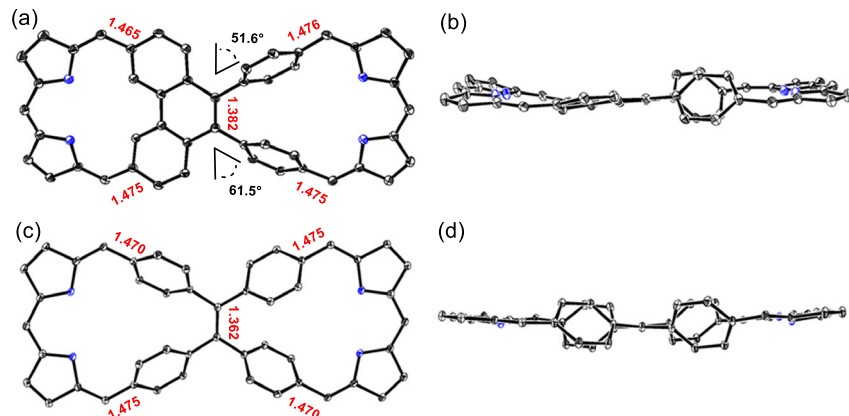

**Fig. 3 | Single-crystal X-ray diffraction structures of 5 and 8. a** Top and (**b**) side views of **5**, (**c**) top and (**d**) side views of **8**. Also shown are the dihedral angles between the two unfused phenylenes and their adjacent benzene rings. Thermal ellipsoids are scaled to the 30% probability level. Hydrogen atoms and *meso*-aryl groups are omitted for clarity.

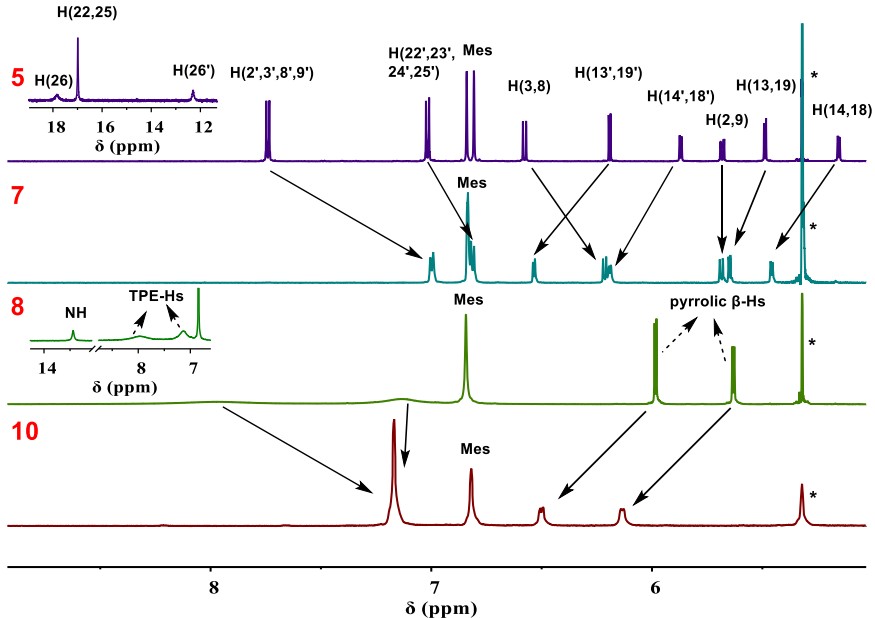

**Fig. 4 | ¹H NMR spectra of 5, 7, 8 and 10.** Partial ¹H NMR spectra of **5**, **7**, **8** and **10** recorded in CD₂Cl₂. Asterisks indicate residual solvent peaks.

the NH signal at the unfused side of **5**. The signals for the pyrrolic β-Hs were observed at 5.99 and 5.64 ppm. Thus, the average chemical shift difference between the pyrrolic NH and β-H protons (Δδ) is 7.63 ppm, a value comparable to what was observed at the unfused side of **5**. We thus suggest that both sides of the symmetric system **8** possess non-aromatic character. We also note that the phenylene protons on the TPE moiety in **8** give rise to broad signals at 7.95 and 7.14 ppm, probably due to the rotation of the phenyl rings. In contrast, the two signals of the unfused phenylene rings on Janus dimer **5** are well-resolved (at 7.74 and 7.02 ppm), reflecting a more rigid structure than **8**.

The UV−vis−NIR absorption spectra of compounds **5** and **8** were systematically recorded in toluene, as illustrated in Fig. 5. Both compounds displayed analogous spectral characteristics, featuring a prominent band around 380 nm and faint absorption signals extending beyond 530 nm. However, distinctive variations were discerned in the corresponding near-infrared (NIR) spectra. These disparities are attributed to alterations in the electronic structure and frontier molecular orbitals (FMOs) resulting from the reduction in symmetry associated with the Janus structure of **5**.

Specifically, compound **8** manifested a peak at 670 nm, characterized by a broad feature attributable to the relatively flexible structure inherent to **8**. Conversely, compound **5** exhibited a red-shifted absorption feature that extended into the NIR region. Such features, indicative of antiaromatic porphyrinoids, are interpreted in the context of optically-forbidden transitions between the highest occupied molecular orbital (HOMO) and lowest unoccupied molecular orbital (LUMO) states. Importantly, in the case of compound **5**, this transition involves a notably reduced energy gap, underscoring its unique electronic properties[61,62].

Discrepancies in the electronic structures of compounds **5** and **8** were further elucidated through their respective excited state dynamics, as depicted in Fig. 6 and Supplementary Fig. 3. Compound **8** exhibited pronounced double exponential decay features in its femtosecond transient absorption (TA) spectrum, characterized by ultrafast decay components of 0.2 and 4 ps. This rapid decay behavior is attributed to the nonaromatic and flexible nature of **8**, which facilitates swift conformational changes in its excited states and results in notably short excited-state lifetimes.

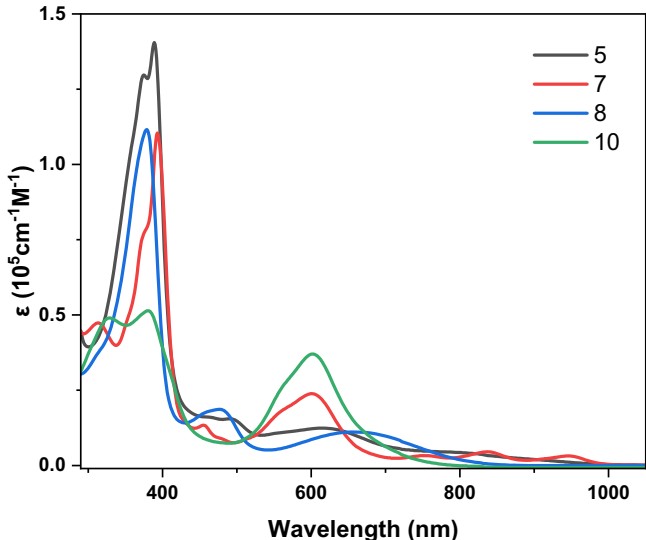

**Fig. 5 | Absorption spectra of 5, 7, 8 and 10.** UV–vis–NIR absorption spectra of **5** (black), **7** (red), **8** (blue), and **10** (green) recorded in toluene.

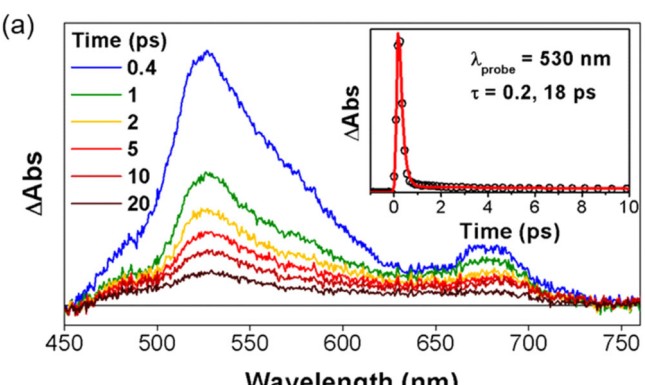

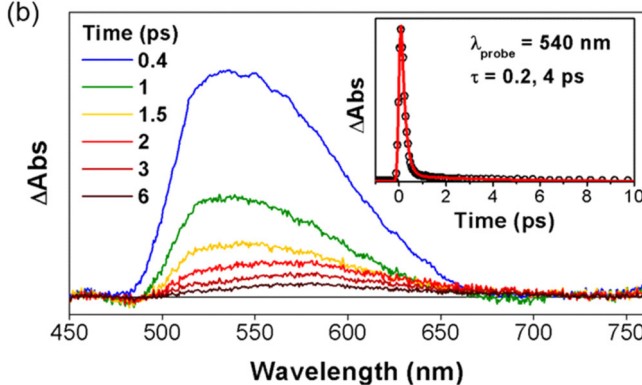

**Fig. 6 | Excited state dynamics of 5 and 8.** TA spectra and decay profiles (insert) of (**a**) **5** and (**b**) **8** in toluene with photoexcitation at 600 nm.

In contrast, transient absorption analyses of compound **5** unveiled a dynamic interplay of fast and relatively slow decay components (0.2 and 18 ps, respectively). The protracted lifetime associated with the 18 ps decay in compound **5** is ascribed to an optically-forbidden transition occurring between the HOMO and LUMO states. This extended decay provides additional evidence supporting the proposed antiaromatic nature of compound **5** while underscoring its unique electronic characteristics[61,62].

The elucidation of the distinctive electronic structures of compounds **5** and **8** was further advanced through the application of time-dependent density functional theory (TD-DFT), as depicted in Supplementary Fig. 6. Compound **8** manifested symmetric electron density distributions, whereas compound **5** exhibited pronounced asymmetry in its electron distribution. Specifically, in the HOMO of pseudo-dimer **5**, electron density was observed to be more delocalized on the fused side. This asymmetry became more apparent in the unoccupied molecular orbitals (LUMO - L + 3). Such asymmetry is thought to play a pivotal role in the red-shifted and attenuated nature of the lowest electronic transition observed in compound **5** as compared to its symmetric counterpart, compound **8** (refer to Supplementary Table 1).

The Janus-like character of compound **5** was further explored through an examination of the electron density of delocalized bonds (EDDB). This computational method quantifies cyclic π-delocalization across conjugated bonds, visually depicting the extent of delocalization or localization within a given molecular framework[63,64]. In the EDDB plot of compound **8**, localized π-conjugation on the unfused phenylene segments is seen that disrupts the π-conjugation throughout the entire carbaporphyrin structure (Fig. 7a, b). In contrast, the presence of the mono-fused phenylene (phenanthrene) subunit in compound **5** facilitates π-delocalization along the fused carbaporphyrin moiety, aligning well with the observed asymmetric distribution of the electron density inferred from the analysis of frontier molecular orbitals (FMOs).

The antiaromatic and nonaromatic character of compounds **5** and **8**, respectively, were further assessed using anisotropy of the induced current density (ACID) and nucleus-independent chemical shift (NICS) calculations. These calculations allow visualization of ring currents and provide estimations of the chemical shifts induced by an external magnetic field, respectively[65,66]. Compound **8** displayed interrupted current surfaces without pronounced current flow in the ACID plot, while compound **5** exhibited a clear antiaromatic counter-clockwise ring current on the fused part, a finding taken as further support for is proposed asymmetric Janus-like character (Supplementary Fig. 7). The NICS(1.5) analysis revealed positive values (a hallmark of antiaromaticity) only within the fused inner cavity of compound **5** (Fig. 7c, d).

Carbaporphyrinoids with adj-CCNN cores are known to coordinate a variety of metal ions, such as Cu(III)[38,58,59], Ag(III)[67], and Pd(II)[38,39], as well as stabilizing hypervalent organophosphorus(V)[57] complexes. The fused side of **5** has a CCNN core and was thus considered potentially capable of stabilizing metal centers in relatively high oxidation states. Conversely, the unfused half was deemed less likely to act as an effective ligand for metal complexes and thus was expected to act more like a classic dipyrromethene unit. To test these expectations, pseudo-dimer **5** was treated with excess $BF_3 \cdot OEt_2$ in toluene in the presence of diisopropylethylamine (DIPEA). Only a mono-BODIPY complex **6** was obtained with the $BF_2$ unit found exclusively within the unfused cavity. This mono-BODIPY complex (**6**) could be further reacted with $Cu(OAc)_2$ in $CHCl_3/CH_3CN$ to give the hetero-complex **7** (Fig. 2a). In contrast, the TPE-linked bis-dibenzihomoporphyrin **8** could be converted fully to the corresponding bis-BODIPY complex **10** by treating with an excess of $BF_3 \cdot OEt_2$ (Fig. 2b). Complexes **7** and **10** were characterized by [1]H, [19]F, 2D-COSY NMR spectroscopies, and UV-vis-NIR spectroscopies, high-resolution mass spectrometry, as well as single crystal X-ray diffraction analyses.

Single crystals of **7** were obtained by the slow diffusion of methanol into an initial toluene solution (Fig. 8a, b). The copper atom within the dicarbacorrole cavity are coordinated to the CCNN coordination environment present on the fused side, resulting in a nearly planar structure in this region. The metal–C and metal–N bond lengths are 1.942, 1.983, 1.921 and 1.883 Å respectively, values that are closely aligned with those reported for various Cu(III) dicarbaporphyrinoids[58,59,68]. Considering its square-planar coordination geometry and in analogy to these prior reports, the oxidation state

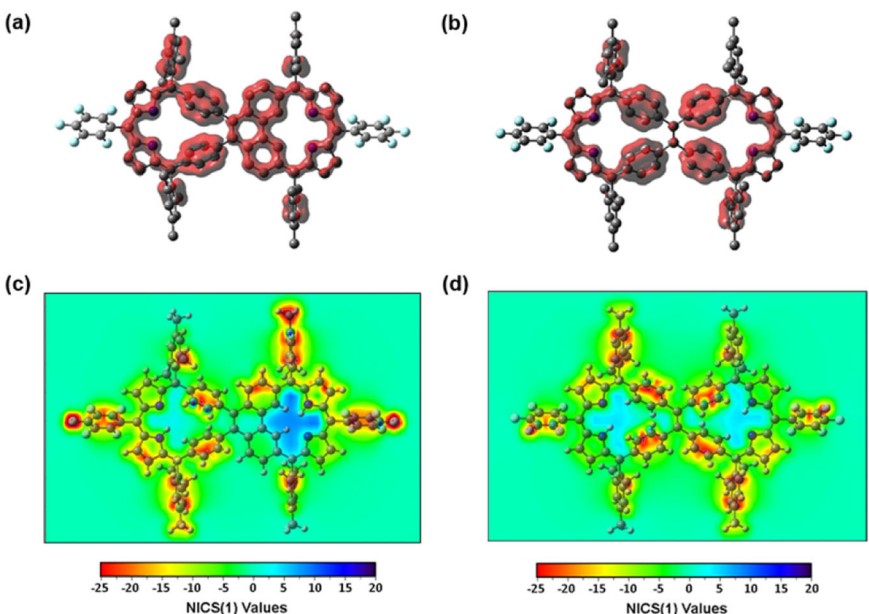

**Fig. 7 | EDDB plots and NICS 2D maps of 5 and 8.** EDDB plots of (**a**) **5** and (**b**) **8**, where the localized and delocalized cyclic π-conjugation pathways are visualized with red-colored surface with an isovalue of 0.014. NICS 2D maps of (**c**) **5** and (**d**) **8**, which are estimated 1.5 Å above their mean plane.

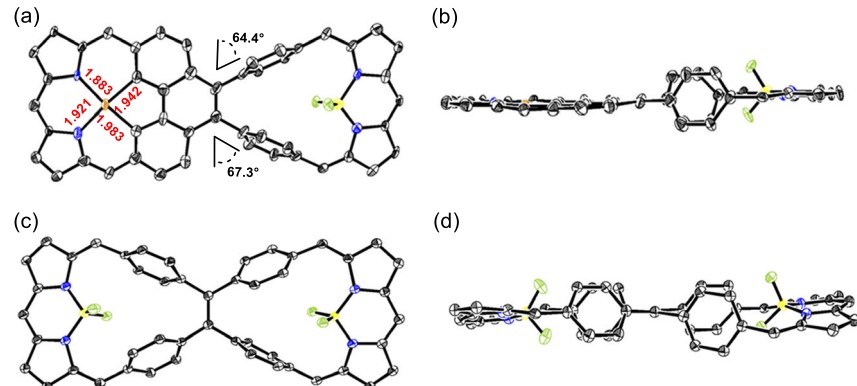

**Fig. 8 | Single-crystal X-ray diffraction structures of complexes 7 and 10. a** Top and (**b**) side views of **7**; (**c**) top and (**d**) side views of **10**. Thermal ellipsoids are scaled to the 30% probability level. Hydrogen atoms and *meso*-aryl groups are omitted for clarity.

of the copper ion in **5** was assigned as +3 (d[8]). On the unfused side, the BF₂ group is bound to the two N atoms of the dipyrromethene unit in classic BODIPY fashion. Following BF₂ complexation, the two unfused phenylenes rotate, presumably to reduce steric hindrance, as evidenced by an increase in the dihedral angles between the phenylenes and the central benzene to 67.3° and 64.4° (compared to 61.5° and 51.6° in **5**), respectively. As expected, the crystal structure of **10** (Fig. 8c, d) revealed a highly symmetric bis-BF₂ coordinated complex, in which the BF₂ groups are coordinated to the dipyrromethene unit, again in BODIPY-like fashion.

The ¹H NMR spectrum of **7** (Fig. 4) provided further evidence for the formation of a diamagnetic organometallic Cu(III) complex. For instance, the NH (H26) and inner CH (H22, 25) signals on the fused side, as well as NH (H26′) signal on the unfused side, all disappeared after coordination with BF₂ and Cu(III). Similarly, the ¹H NMR spectra of the bis-BF₂ complex **10** (Fig. 4) revealed the absence of discernible NH signals, which were clearly evident in the case of the starting free-base **8**. The signals of the phenylenes, originally present in the form of two sets of broad peaks in **8**, were present as one sharp set of peaks in the case of **10**. This finding is interpreted in terms of BF₂ coordination not

only occurring as suggested, but also inducing increased structural rigidity. Furthermore, ¹⁹F NMR spectral analyses of **7** and **10** revealed signals ascribable to the BF₂ moieties at ca. −123 and −120 ppm, respectively, as well as separate signals for the pentafluorophenyl substituents (Supplementary Figs. 31 and 40).

The UV-Vis-NIR absorption spectrum of complex **7** revealed the presence of weak absorption bands ≥700 nm that were slightly red-shifted and better resolved those seen in the spectrum of the free-base form **5** (Fig. 5). These spectral features of **7** recapitulate those seen previously in the case of an antiaromatic bis-Cu(III) complex of the control bis-dicarbacorrole **9**[38]. We thus propose that an enhancement in the overall π-conjugation is induced by metalation, which, per the structural analysis, enforces a near-planar geometry in the fused portion of **7**. A prominent and intense absorption band at 603 nm was also observed in the absorption spectra of both **7** and **10**. Complex **10** displayed a higher extinction coefficient for this band compared to **7**. Moreover, significantly attenuated and split bands were seen at 330 and 382 nm (Fig. 5). These features closely resemble those reported for a nonaromatic macrocycle possessing analogous BODIPY units[69]. Thus, taken in concert, these findings provide

support for the suggestion that 7 represents a true hetero-complex wherein two very different metal(oid)s, namely Cu(III) and $BF_2$, are bound.

The excited-state dynamics of 7 and 10, as observed by TA measurements, reveal their distinct electronic structures (Supplementary Fig. 4). Compound 7 exhibited a fast single TA spectral decay with a time constant of 3 ps, while its TA spectral shape remained analogous to that of 5. This leads us to propose that photoexcitation initially excites the π-conjugated ligand, and an effective metal-ligand interaction through the planar structure of 7 leads to a subsequent ligand-to-metal charge transfer[70,71]. The spectral similarity between 7 and 5 reflects a resemblance in their π-electronic structures, thus providing support for the notion that complex 7 retains the antiaromatic features seen in the free-base form. In marked contrast, the TA spectra of 10 were characterized by double exponential decays with time constants of 1.5 and 21.5 ps, respectively. A significant spectral evolution was observed during the former time constant period, a finding we interpret in terms of a large conformational change occurring in the excited state, as commonly observed in large nonaromatic macrocycles[59,72,73].

The effect of Cu- and $BF_2$-coordination on 5 and 8 was further analyzed by means of EDDB, ACID, and NICS calculations for 7 and 10. The EDDB plots proved very similar π-electronic structures of 7 and 10 to those of 5 and 8, respectively (Supplementary Fig. 8). The enhanced π-conjugative interaction of 7 by the Cu- and B-coordination resulted in an obvious π-delocalization on the carbaporphyrin unit. This stands in contrast with the individually separated π-conjugation in 10. In line with the EDDB results, the ACID plot of 10 revealed no specific current flow and the presence of disconnected current surfaces; this is as would be expected for a nonaromatic system (Supplementary Fig. 9). In contrast, the expected antiaromatic (paratropic) ring current for the fused portion of 7 was seen in its ACID plot. Positive NICS(1.5) values were also seen around the Cu atom of 7, findings taken as evidence for antiaromatic character persisting within the fused carbaporphyrin portion of 7 (Supplementary Fig. 8).

The redox properties of 5, 7, 8 and 10 were investigated by means of cyclic voltammetry (CV) and differential pulse voltammetry (DPV) carried out in $CH_2Cl_2$ containing 0.1 M $NBu_4PF_6$ as the electrolyte (Supplementary Fig. 5). The first oxidation and reduction potentials of 5 were found to be 0.28 and −1.41 V (vs $Fc^+$/Fc), respectively, resulting in an electrochemical HOMO − LUMO gap of 1.69 eV. Similarly, the HOMO-LUMO gap of 7 was measured as 1.71 eV, with the first oxidation and reduction potentials being observed 0.32 and −1.39 V, respectively. Compound 8 exhibited an oxidation wave at 0.22 V, a value roughly comparable to that seen in the case of 5. However, its first reduction wave was cathodically shifted by 0.30 V (to −1.71 V), a finding interpreted in terms of fusion of the phenylenes making the macrocycle easier to reduce. After $BF_2$ coordination, the HOMO-LUMO gap of 10 increased slightly to 2.00 eV, as compared to the value of 1.93 eV seen for 8. This finding is consistent with the blue shift seen in the absorption bands. Supporting time-dependent density functional theory (TD-DFT) calculations provided estimated HOMO − LUMO energy gaps of 1.852, 1.849, 1.935 and 2.039 eV for 5, 7, 8, and 10, respectively. These values are fully consistent with the experimentally derived redox potentials.

## Discussion

In conclusion, the carbaporphyrin pseudo-dimer 5 is characterized by readily discernible Janus features, with one side exhibiting antiaromatic characteristics and the other side displaying nonaromatic properties. The Janus nature of this system was confirmed through comprehensive analyses, including [1]H NMR spectroscopic studies, single crystal X-ray diffraction analysis, and density functional theory (DFT) calculations. A symmetric control, the TPE-linked nonaromatic carbaporphyrin dimer 8, was also prepared. Systematic comparisons of the photophysical and coordination properties of 5, 8, and a previously reported symmetric antiaromatic carbaporphyrin dimer, 9, demonstrated Janus-like attributes not only in the electronic structure of 5, but also in its complexation chemistry. Notably, compound 5 exhibited unique reactivity, supporting the formation of a mixed Cu(III)-$BF_2$ complex, distinguishing it from the pure bis-$BF_2$ and bis-Cu(III) complexes stabilized by 8 and 9, respectively. This study underscores the advantages of breaking the inherent symmetry of carbaporphyrin dimers, paving the way for applications unattainable in more homogeneous coordination or electronic environments.

## Methods

### Materials and characterization

Chemicals were purchased from commercial sources and used without further purification unless otherwise indicated. Analytical grade solvents, including dichloromethane (DCM), n-hexane, ethyl acetate (EA), methanol (MeOH), tetrahydrofuran (THF), petroleum ether (PE) and toluene were purchased from Beijing Tong Guang Fine Chemicals Company. DCM, when used as a solvent, was distilled over $CaH_2$ under nitrogen. NMR spectra ([1]H NMR spectra at 400 MHz or 600 MHz, [13]C NMR spectra at 100 MHz or 150 MHz, [19]F NMR spectra at 376 or 565 MHz, two-dimensional NMR spectra at 400 or 600 MHz) were recorded on a JEOL NMR spectrometer. High resolution ESI and APCI mass spectra were obtained using Agilent QTOF 6550 or Shimadzu LCMS-IT-TOF instruments. High resolution MALDI-TOF mass spectrometric analyses were carried out using a Bruker Autoflex speed mass spectrometer. UV-vis spectra were measured on a UV-3600i Plus spectrometer from Shimadzu. Recycling preparative gel permeation chromatography (GPC) was performed with a LC-5060 P2 instrument from Japan Analytical Industry Co., Ltd. Cyclic voltammetry (CV) and differential pulse voltammetry (DPV) studies were carried out on a CH Instrument Model 660E electrochemical system utilizing a three-electrode configuration consisting of a glassy carbon (working electrode), platinum wire (counter electrode) and SCE (saturated calomel electrode; reference electrode) in a solution of 0.1 M $TBAPF_6$ in dry DCM at a scan rate of 50 mV s$^{-1}$ in a nitrogen-filled cell. Crystallographic data were collected using a Rigaku Oxford Diffraction Xta-LAB Synergy diffractometer equipped with a HyPix-6000E area detector at 100 K using Cu Kα (λ = 1.54184 Å) from a PhotonJet microfocus X-ray source. Femtosecond transient absorption data were obtained using a lab-built femtosecond absorption spectrometer[74]. After the measurements, the absorption spectra were carefully checked to detect if there were artifacts due to degradation and photooxidation of samples.

### Reporting summary

Further information on research design is available in the Nature Portfolio Reporting Summary linked to this article.

## Data availability

The authors declare that the data supporting the findings of this study are available within the article and Supplementary Information file, or from the corresponding author upon request. The X-ray crystallographic data of corresponding structures reported in this study have been deposited to the Cambridge Crystallographic Data Center (CCDC), under deposition numbers 2286144, 2286143, 2286146, 2286274 (5, 7, 8, 10). These data can be obtained free of charge from CCDC via www.ccdc.cam.ac.uk/data_request/cif. All the other supplementary data are available from the article and its Supplementary Information files or available from the corresponding authors upon request.

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

## Acknowledgements

The work at Beijing Normal University (BNU) was supported by the startup funding (grant no. 312232114 to X.-S.K.). The work at Yonsei University was supported by the National Research Foundation of Korea (NRF) funded by the Korea Government (MSIT) (No. 2020R1A5A1019141 to D.K.). The work at Soonchunhyang University was supported by the NRF funded by the Korea Government (MSIT) (No. 2021R1A6A1A03039503 to J.O.) and Soonchunhyang University Research Fund. The quantum mechanical calculations were performed using the supercomputing resources of the Korea Institute of Science and Technology Information. The work in Austin was supported by the Robert A. Welch Foundation (F-0018 to J.L.S.) and by the National Science Foundation (CHE-2304731 to J.L.S.) subsequent to Nov. 1, 2023.

## Author contributions

X.-S.K. conceived the project. H.H. performed the compound syntheses and characterization, grew the single crystals, measured the absorption spectra and conducted the electrochemical studies. J.L. and J.K. carried out the theoretical calculations and fs-TA measurements. Z.Z. and V.M.L. performed X-ray crystallography. J.O., D.K., J.L.S. and X.-S.K. coordi-nated and supervised the project. All authors contributed to the manuscript.

## Competing interests

The authors declare no competing interests.
