## [Peer Review File · Nature Communications]

REVIEWER COMMENTS

Reviewer #1 (Remarks to the Author):

The manuscript "A Janus Carbaporphyrin Pseudo-Dimer" submitted by He et al. is an exceptional contribution to carbaporphyrinoid chemistry. Once again, the joint effort of these outstanding laboratories resulted in preparing and characterizing novel porphyrinoids, in this case unusual "pseudo-dimeric" carbaporphyrinoids. It is quite amazing to see what a phenanthrene-like moiety will do once incorporated into an appropriate porphyrinoid environment. The authors have carefully characterized the new compounds with the usual techniques (UV-vis-NIR absorption spectra spectroscopy, femtosecond transient absorption (TA) spectra, HRMS) and presented crystallographic structures for the title compound 5 and its mixed copper(III) – boron(III) complex. When dealing with porphyrinoids, the questions of their magnetic behavior must be dealt with - this was done via convincing ¹H NMR experiments. Significantly, the detailed density functional theory (DFT) calculations afforded the appropriate insight into the electronic structure. In particular, one can appreciate the detailed examination of the electron density of delocalized bonds (EDDB), and NICS 2D maps used to address the antiaromatic or nonaromatic character of the relevant Janus carbaporphyrin pseudo-dimer as nicely shown in Figure 5. The induced current density (ACID) calculations have been reported as well. Remarkably, the Janus "pseudo-dimer" contains two distinct carbaporphyrinoid-like pockets. The molecule displays antiaromatic features on the phenanthriporphyrin-like side and nonaromatic behavior on the unfused side. One can also appreciate the different coordinating properties of both subunits which opens the nontrivial way for future exploration applying the control by virtue of the differences in coordinating properties of each subunit. Concluding, this is a highly important contribution to the field of carbaporphyrinoids in general and specifically to aceneporphyrinoids. The Janus-like coordination motive, imposed by the specific molecular structure, is highly appreciated to be of use in creative coordination/catalytic chemistry or materials science.

Thus I very strongly support the publication of this contribution in Nature Communication, however, after some modifications outlined below.

A.

In Introduction authors presented a detailed presentation of the state of the art in the field, Still the Review by Szyszko should be mentioned as the relevant reference. as well "Phenanthrene-Embedded Carbaporphyrinoids and Related Systems: From Ligands to Cages and Molecular Switches Eur. J. Org. Chem. 2022, e202200714

B

The discussion on the dimeric symmetric carbaporphyrinoids was concluded as follows.

"Despite these achievements, the exploration of carbaporphyrin dimers with two distinct porphyrin-like subunits remains essentially uncharted territory"

In light of this particular comment, I would like to draw the attention of the authors to the fact that this uncharted territory has some rare inhabitants. One can notice three relevant reports. Importantly, the carbaporphyrin dyads (pseudo-dimers) with two distinct porphyrin-like subunits have been originally reported in these contributions.

1.

The peculiar asymmetric carbathiaporphyrin–carbathiachlorin dyad (compound 8 in ref *Angew. Chem. Int. Ed.* 2016, 55, 11231, DOI DOI: 10.1002/anie.201606298, ref. 29 in the reviewed manuscript work)

2

21-Carba-23-Selenaporphyrinoid dyads characterized by a canted arrangement of two carbaselenaporphyrinic planes. (*Angew. Chem. Int. Ed.* 2023, 62, e202314925. doi.org/10.1002/anie.202314925)

These dyads exhibit a novel geometry with two structurally different subunits. All heterodimers have an aromatic 21-carba-23-selenaporphyrin component accompanied alternatively by aromatic or nonaromatic subunits.

3 ‘

Triphyrin(2.1.1) motif built on the extended pi-system of a tetraphyrin(1.1.1.1) skeleton afforded an additional coordination environment, with a CNN set of donors confined in a limited space. *Angew. Chem. Int. Ed.* . 2019, 58, 10946 DOI: 10.1002/anie.201904819.

C.

The comparison of the chemical shifts of 5 to the free phenathriporphyrin, instead of those of 9, should be included in the discussion. In fact, 5 contains a regular phenathriporphyrin subunit which is electronically different from the subunit of 9. The interaction between two competing moieties in 9 can significantly influence the chemical shifts.

Reviewer #2 (Remarks to the Author):

Carbaporphyrin-type systems were first reported 30 years ago and have become increasingly investigated due to their unique properties. In this paper the authors have investigated a unique pseudo-dimer that can be considered to be a Janus-type species due to the presence of two faces or at least two connected cavities. Symmetrical carbaporphyrin dimers have been reported but asymmetrical systems are not well known. A tetrakis(bromophenyl)ethylene undergoes oxidative cyclization to give a diarylphenanthrene.

Following metal-halogen exchange and addition of mesitaldehyde, a tetra-alcohol is generated. Reaction with excess pyrrole in the presence of boron trifluoride etherate and cyclization with pentafluorobenzaldehyde afforded the so-called pseudo-dimer. The system has two fused porphyrinoid units; one is antiaromatic while the other is nonaromatic. The nonaromatic unit readily forms a BF₂ complex but not the antiaromatic component. The latter gave the copper(III) complex and this opens up the construction of bimetallic species. Symmetrical structures do not show this type of property. The new porphyrinoids were thoroughly characterized using spectroscopic methods. X-ray structures are also provided. Photophysical and electrochemical studies were performed and the electronic spectra were analyzed. Several computational methods were used, in conjunction with the proton NMR data, to assess the anti/nonaromatic character, specifically the electron density of delocalized bonds (EDDB), NICS calculations and ACID plots. This is a very thorough and high quality study. The paper is well written and I have only a few minor points to make, none of which are very important.

1. A carbaporphyrin is a porphyrin-type structure with a carbon in place of one of the nitrogens. Should these be considered to be carbaporphyrins or carbaporphyrin-like species? i.e. carbaporphyrinoids. At the top of column 2 on page 5, "which is sharp contrasting" might be written as "which sharply contrasts with the interrupted pi-conjugation in 10. In the following sentence "this is as would be expected".

3. Although well written, the introduction described a lot of information that appears in the results section. It seems a little long.

This is a really nice piece of work and I support publication.

Reviewer #3 (Remarks to the Author):

The authors mainly report the synthesis of a novel Janus-type carbaporphyrin pseudo-dimer 5 and its properties. All the properties of 5 and comparison studies of 8 support that pseudo-dimer 5 displays antiaromatic characteristics on the fused side and nonaromatic behavior on the unfused side. The Janus dimer also shows regioselectivity for Cu and BF₃. These properties turn pseudo-dimer 5 into a heterometallic complex with Cu and BF₂.

These two distinct properties in one porphyrinoid are somewhat expected. However, this reviewer agrees that 'the exploration of carbaporphyrin dimers with two distinct porphyrin-like subunits remains essentially uncharted territory,' as the authors stated. Thus, this manuscript can be acceptable after the suggested corrections below.

minor issues

-Page 1 "with disparate aromaticity feature"

It makes reader confuse whether pseudo-dimer 5 has two different levels of aromaticity.

-Page 3 "the two unfused"

According to this reviewers' understanding, "the two unfused phenylene rings on Janus dimer 5 shows" should be "the two signal of the unfused~."

-Page 3

Compound 10 first appears without its structural information in Fig 3. This reviewer does not know what it is until seeing fig 7. There are many compound numbers to remember to understand this manuscript. Authors may need more simple numbering system for readers. Compound numbers bother reader a lot. There is another level of numbering issue. Numbers should appear in the order in Fig 1.

Reference 33

authors should fix inconsistent font size of reference 33.

Reply to Reviewers' Comments:

< Response to Reviewer 1 >

Reviewer #1

Remarks to the Author:

The manuscript "A Janus Carbaporphyrin Pseudo-Dimer " submitted by He et al. is an exceptional contribution to carbaporphyrinoid chemistry. Once again, the joint effort of these outstanding laboratories resulted in preparing and characterizing novel porphyrinoids, in this case unusual "pseudo-dimeric" carbaporphyrinoids. It is quite amazing to see what a phenanthrene-like moiety will do once incorporated into an appropriate porphyrinoid environment. The authors have carefully characterized the new compounds with the usual techniques (UV-vis-NIR absorption spectra spectroscopy, femtosecond transient absorption (TA) spectra, HRMS) and presented crystallographic structures for the title compound 5 and its mixed copper(III) – boron(III) complex. When dealing with porphyrinoids, the questions of their magnetic behavior must be dealt with - this was done via convincing ¹H NMR experiments. Significantly, the detailed density functional theory (DFT) calculations afforded the appropriate insight into the electronic structure. In particular, one can appreciate the detailed examination of the electron density of delocalized bonds (EDDB). and NICS 2D maps used to address the antiaromatic or nonaromatic character of the relevant Janus carbaporphyrin pseudo-dimer as nicely shown in Figure 5. The induced current density (ACID) calculations have been reported as well. Remarkably, the Janus "pseudo-dimer" contains two distinct carbaporphyrinoid-like pockets. The molecule displays antiaromatic features on the phenanthriporphyrin-like side and nonaromatic behavior on the unfused side. One can also appreciate the different coordinating properties of both subunits which opens the nontrivial way for future exploration applying the control by virtue of the differences in coordinating properties of each subunit. Concluding, this is a highly important contribution to the field of carbaporphyrinoids in general and specifically to aceneporphyrinoids. The Janus-like coordination motive, imposed by the specific molecular structure, is highly appreciated to be of use in creative coordination/catalytic chemistry or materials science. Thus I very strongly support the publication of this contribution in Nature Communication, however, after some modifications outlined below.

Response: We appreciate this reviewer's positive comments on this work and their recommendation to publish this manuscript in *Nature Commun.*

A.

In Introduction authors presented a detailed presentation of the state of the art in the field, Still the Review by Szyszko should be mentioned as the relevant reference. as well "Phenanthrene-Embedded Carbaporphyrinoids and Related Systems: From Ligands to Cages and Molecular Switches *Eur. J. Org. Chem.* 2022, e202200714

Response: Thanks for the suggestions. This reference has been cited in the revised manuscript.

B.

The discussion on the dimeric symmetric carbaporphyrinoids was concluded as follows.

“Despite these achievements, the exploration of carbaporphyrin dimers with two distinct porphyrin-like subunits remains essentially uncharted territory”

In light of this particular comment, I would like to draw the attention of the authors to the fact that this uncharted territory has some rare inhabitants. One can notice three relevant reports. Importantly, the carbaporphyrin dyads (pseudo-dimers) with two distinct porphyrin-like subunits have been originally reported in these contributions.

1.

The peculiar asymmetric carbathiaporphyrin–carbathiachlorin dyad (compound 8 in ref Angew. Chem. Int. Ed. 2016, 55, 11231, DOI DOI: 10.1002/anie.201606298, ref. 29 in the reviewed manuscript work)

2

21-Carba-23-Selenaporphyrinoid dyads characterized by a canted arrangement of two carbaselenaporphyrinic planes. (Angew. Chem. Int. Ed. 2023, 62, e202314925. doi.org/10.1002/anie.202314925)

These dyads exhibit a novel geometry with two structurally different subunits. All heterodimers have an aromatic 21-carba-23-selenaporphyrin component accompanied alternatively by aromatic or nonaromatic subunits.

3

Triphyrin(2.1.1) motif built on the extended pi-system of a tetraphyrin(1.1.1.1) skeleton afforded an additional coordination environment, with a CNN set of donors confined in a limited space. Angew. Chem. Int. Ed. 2019, 58, 10946 DOI: 10.1002/anie.201904819.

Response: Thanks for these constructive suggestions. Indeed, compound 8 in Angew. Chem. Int. Ed. 2016, 55, 11231 represents an asymmetric carbathiaporphyrin–carbathiachlorin dyad. We re-cited this paper in the revised manuscript. The two other papers called to our attention are now cited in the revised manuscript.

C.

The comparison of the chemical shifts of 5 to the free phenathriporphyrin, instead of those of 9, should be included in the discussion. In fact, 5 contains a regular phenathriporphyrin subunit which is electronically different from the subunit of 9. The interaction between two competing moieties in 9 can significantly influence the chemical shifts.

Response: Thanks for the suggestion. A comparison of the chemical shifts of 5 vs those of the free base phenathriporphyrin is included in the revised manuscript.

< Response to Reviewer 2 >

Reviewer #2

Remarks to the Author:

Carbaporphyrin-type systems were first reported 30 years ago and have become increasingly investigated due to their unique properties. In this paper the authors have

investigate a unique pseudo-dimer that can be considered to be a Janus-type species due to the presence of two faces or at least two connected cavities. Symmetrical carbaporphyrin dimers have been reported but asymmetrical systems are not well known. A tetrakis(bromophenyl)ethylene undergoes oxidative cyclization to give a diarylphenanthrene. Following metal-halogen exchange and addition of mesitaldehyde, a tetra-alcohol is generated. Reaction with excess pyrrole in the presence of boron trifluoride etherate and cyclization with pentafluorobenzaldehyde afforded the so-called pseudo-dimer. The system has two fused porphyrinoid units; one is antiaromatic while the other is nonaromatic. The nonaromatic unit readily forms a BF₂ complex but not the antiaromatic component. The latter gave the copper(III) complex and this opens up the construction of bimetallic species. Symmetrical structures do not show this type of property. The new porphyrinoids were thoroughly characterized using spectroscopic methods. X-ray structures are also provided. Photophysical and electrochemical studies were performed and the electronic spectra were analyzed. Several computational methods were used, in conjunction with the proton NMR data, to assess the anti/nonaromatic character, specifically the electron density of delocalized bonds (EDDB), NICS calculations and ACID plots. This is a very thorough and high quality study. The paper is well written and I have only a few minor points to make, none of which are very important.

Response: We appreciate this reviewer's supportive evaluation.

1. A carbaporphyrin is a porphyrin-type structure with a carbon in place of one of the nitrogens. Should these be considered to be carbaporphyrins or carbaporphyrin-like species? i.e. carbaporphyrinoids. At the top of column 2 on page 5, "which is sharp contrasting" might be written as "which sharply contrasts with the interrupted pi-conjugation in 10. In the following sentence "this is as would be expected".

Response: Thanks for the thoughtful suggestions. Carbaporphyrin with a built-in cyclopentadienyl moiety has been called a "true carbaporphyrin" (Latos-Grażyński, et al. *Angew. Chem. Int. Ed.* **2014**, 53, 4885.) However, more broadly, porphyrinoid systems wherein a pyrrole moiety is replaced by an arene are now called carbaporphyrin or carbaporphyrinoid by most members of the community. We feel both terms should be useful in describing the lead compounds in this manuscript in a way that is accessible to a broad readership.

In addition, the other typographical error-type issues have been corrected in the revised manuscript.

3. Although well written, the introduction described a lot of information that appears in the results section. It seems a little long.

This is a really nice piece of work and I support publication.

Response: Thanks for the suggestions. We have now deleted the paragraph in the introduction that describes Janus systems in general. Indeed, as this reviewer notes, these aspects are mentioned in the results section.

< Response to Reviewer 3 >

Reviewer #3

Remarks to the Author:

The authors mainly report the synthesis of a novel Janus-type carbaporphyrin pseudo-dimer 5 and its properties. All the properties of 5 and comparison studies of 8 support that pseudo-dimer 5 displays antiaromatic characteristics on the fused side and nonaromatic behavior on the unfused side. The Janus dimer also shows regioselectivity for Cu and BF₃. These properties turn pseudo-dimer 5 into a heterometallic complex with Cu and BF₂.

These two distinct properties in one porphyrinoid are somewhat expected. However, this reviewer agrees that ‘the exploration of carbaporphyrin dimers with two distinct porphyrin-like subunits remains essentially uncharted territory,’ as the authors stated. Thus, this manuscript can be acceptable after the suggested corrections below.

Response: We appreciate this reviewer’s kind evaluation of this work and endorsement of publication in *Nature Commun.*

Minor issues

-Page 1 “with disparate aromaticity feature”

It makes reader confuse whether pseudo-dimer 5 has two different levels of aromaticity.

Response: Thanks for the suggestion. We modified the description in the revised manuscript.

Page 3 “the two unfused”

According to this reviewers’ understanding, “the two unfused phenylene rings on Janus dimer 5 shows” should be “the two signal of the unfused~.”

Response: Thanks for the suggestion. We modified the description in the revised manuscript.

-Page 3

Compound 10 first appears without its structural information in Fig 3. This reviewer does not know what it is until seeing fig 7. There are many compound numbers to remember to understand this manuscript. Authors may need more simple numbering system for readers. Compound numbers bother reader a lot. There is another level of numbering issue. Numbers should appear in the order in Fig 1.

Response: Thanks for the kind suggestion. We modified Fig. 1 in the revised manuscript. In addition, we added a Scheme 1b that shows the synthesis from the control dimer 8 to corresponding bis-BF₂ complex 10. We hope these changes make the manuscript easier to understand.

Reference 33

authors should fix inconsistent font size of reference 33.

Response: Thanks for the suggestion. The issue has been fixed in the revised manuscript.